# Photophysical Deactivation Mechanisms of the Pyrimidine Analogue 1-Cyclohexyluracil

**DOI:** 10.3390/molecules26175191

**Published:** 2021-08-27

**Authors:** Danillo Valverde, Adalberto V. S. de Araújo, Antonio Carlos Borin

**Affiliations:** Department of Fundamental Chemistry, Institute of Chemistry, University of São Paulo, Avenida Professor Lineu Prestes, 748, São Paulo 05508-000, SP, Brazil; adalberto.araujo@usp.br

**Keywords:** 1-cyclohexyluracil, uracil derivative, photochemical deactivation pathways

## Abstract

The photophysical relaxation mechanisms of 1-cyclohexyluracil, in vacuum and water, were investigated by employing the Multi-State CASPT2 (MS-CASPT2, Multi-State Complete Active-Space Second-Order Perturbation Theory) quantum chemical method and Dunning’s cc-pVDZ basis sets. In both environments, our results suggest that the primary photophysical event is the population of the $S_{1}$${}^{1}\left(\pi \pi^{*}\right)$ bright state. Afterwards, two likely deactivation pathways can take place, which is sustained by linear interpolation in internal coordinates defined via Z-Matrix scans connecting the most important characteristic points. The first one (Route 1) is the same relaxation mechanism observed for uracil, its canonical analogue, i.e., internal conversion to the ground state through an ethylenic-like conical intersection. The other route (Route 2) is the direct population transfer from the $S_{1}$${}^{1}\left(\pi \pi^{*}\right)$ bright state to the $T_{2}$${}^{3}\left(n\pi^{*}\right)$ triplet state via an intersystem crossing process involving the ($S_{1}$${}^{1}\left(\pi \pi^{*}\right)$/$T_{2}$${}^{3}\left(n\pi^{*}\right)$)${}_{STCP}$ singlet-triplet crossing point. As the spin-orbit coupling is not too large in either environment, we propose that most of the electronic population initially on the $S_{1}$${}^{1}\left(\pi \pi^{*}\right)$ state returns to the ground following the same ultrafast deactivation mechanism observed in uracil (Route 1), while a smaller percentage goes to the triplet manifold. The presence of a minimum on the $S_{1}$${}^{1}\left(\pi \pi^{*}\right)$ potential energy hypersurface in water can help to understand why experimentally it is noticed suppression of the triplet states population in polar protic solvent.

## 1. Introduction

The five canonical nucleobases (adenine, guanine, cytosine, thymine, and uracil), which constitute our genetic alphabet, are the building blocks of DNA and RNA, being distinguished by their high photostability due to efficient decay mechanisms involving extremely fast nonradiative deactivation processes via internal conversion (conical intersection) to the ground state, minimizing the effects of undesirable photoexcited chemical reactions [1,2,3,4,5,6,7,8,9]. This outstanding characteristic has implications from theories about the origin of life on our planet [10,11] to pharmaceutic products [12,13], such as sunscreens, that protect our skin from UV radiation.

Among the most common DNA photodamages, there are those related to the cyclobutane pyrimidine dimers (CPDs) and pyrimidine (6–4) pyrimidone photoproducts [14,15,16]. There is also evidence of DNA photodamage triggered by long-lived triplet excited states by means of likely triplet energy transfer mechanisms [16,17,18,19]. Nonetheless, in spite of the substantial knowledge about the mechanisms related to the ultrafast internal processes involving the singlet excited states, the role of the triplet excited states and their intrinsic decay mechanisms, ruled by intersystem crossing (ISC) processes, is much less clear, as has been shown by some previous theoretical studies [8,20].

1-Cyclohexyluracil (1CHU, Figure 1), a uracil derivative with a cyclohexyl group covalently bounded to the $N_{1}$ position, the same to which the sugar moiety is attached in the RNA structure, was the first system whose triplet state population was measured for the first time. The author employed the femtosecond transient absorption spectroscopy (TAS) technique to conclude that the triplet states could be populated in less than 10 ps in protic and aprotic solvents just varying the polarity [21]. Furthermore, the authors mentioned that the triplet state population mechanism would compete with other ultrafast internal conversion processes to the ground-state already observed in RNA pyrimidine monomers, since both take place at the same time scale. These experimental findings were corroborated by high-level ab initio calculations in similar systems [9,22,23]. Not long ago, canonical nucleobases and modified analogues received considerable attention from experimenters with the focus on better understanding their triplet population mechanism [24,25,26,27,28,29].

The 1CHU molecule is a good compound to study triplet formation because it is soluble in solvents covering a wide range of polarity; in addition, the ${}^{1}\left(\pi \pi^{*}\right)$ and ${}^{1}\left(n\pi^{*}\right)$ electronic states can behave differently depending on the polarity. Moreover, the presence of the cyclohexyl substituent in the $N_{1}$ position approximates the tautomeric behavior of this model molecule to that observed in biological environments, the main situation of interest for nucleobases photophysics. New Watson–Crick base pairing [30,31] schemes and specific hydrogen bonding interactions with amino acid in chloroform solution [32] involving this molecule has called the attention of the scientific community. There is still some discussion of an eventual formation of aggregates in chloroform solution [33].

As for experimental results about the excited states of 1CHU, Hare et al. [21] measured the 1CHU triplet population yield ($\Phi_{T}$) in different solvents such as water (ε=80.1, polar and protic solvent), acetonitrile (ε=36.6, polar and aprotic solvent), and ethyl acetate (ε=6.1, apolar and aprotic solvent). Recently, Brister and Crespo-Hernández [34], based on broadband transient absorption spectroscopic results, concluded that the 1CHU triplet manifold is populated in a femtosecond time regime in acetonitrile. In the same paper, they also carried out single-point vertical excitation calculations on the Franck-Condon (FC) region at the B3LYP/6-311++G(d,p) level of theory and the Polarizable Continuum Model (PCM) [35] to simulate acetonitrile solvation effects. According to their results, the triplet state is more accessible in acetonitrile than in vacuum because the energy difference between the ${}^{1}\left(n\pi^{*}\right)$ and ${}^{3}\left(\pi \pi^{*}\right)$ states is smaller in the former. As these states bear distinct electronic configurations, the ISC should be more favorable in conformity with the El-Sayed rule [36]. In addition, 1CHU exhibits a slightly higher $\Phi_{T}$ in comparison to its respective canonical counterpart, namely uracil, in the same conditions [37,38].

To the best of our knowledge, theoretical studies about 1CHU have been planned to describe the nature of the electronic states and related properties on the Franck–Condon region only, which is not enough to understand the mechanisms behind its photophysical properties. Therefore, we decided to carry out a comprehensive investigation of the most relevant relaxation pathways, which will give further support to the previous experimental results [21,34]. To this end, the Multi-State CASPT2 (MS-CASPT2, Multi-State Complete Active-Space Second-Order Perturbation Theory) method and double-ζ basis sets were employed to describe the 1CHU relaxation mechanisms via a systematic description of the most relevant potential energy surfaces (PESs), minimum energy regions, minimum energy and singlet-triplet crossing points, and spin-orbit couplings. Furthermore, a comparison with the well-known photophysics of uracil [8,20,39,40] will be also presented.

## 2. Results and Discussion

### 2.1. Franck–Condon Region and Absorption Spectrum

The gas phase ground state geometry $S_{0}$${}_{\min}$, optimized at the MS(3S)-CASPT2(12,9)/cc-pVDZ level of theory, exhibits a planar pyrimidine ring and the cyclohexyl group with a chair conformation. Our optimized geometry is similar to that reported earlier by Brister and Crespo-Hernández [34], computed at the B3LYP/6-11++G(p,d) level of theory, with the largest difference observed for the $C_{5}C_{6}$ bond, being 0.014 Å longer than that predicted at the MS-CASPT2 level. We can also notice that the 1CHU and uracil ground-state optimized geometries are similar (Table 1).

Vertical excitation energies for the lowest-lying singlet and triplet electronic states computed at the FC region are listed in Table 2. The $S_{1}$${}^{1}\left(\pi_{5}\pi_{6}^{*}\right)$ state is the lowest-lying singlet excited state placed 4.71 eV (μ=7.27 Debye) vertically above the ground state. In relation to the ground state, it is derived from a single excitation from the $\pi_{5}$ orbital to the $\pi_{6}^{*}$ anti-bonding orbital (see Section 3 for details about the active space). Within the experimental excitation window, the transition to the $S_{1}$${}^{1}\left(\pi_{5}\pi_{6}^{*}\right)$ state is predicted to be the most intense, with the largest oscillator strength (f=0.360) computed. Therefore, it will carry most of the population upon irradiation and can be referred to as the bright state. It is worth noting that the computed $S_{1}$${}^{1}\left(\pi_{5}\pi_{6}^{*}\right)$ excitation energy is in agreement with the experimental value (4.68 eV) [34] obtained in ethyl acetate solution, an apolar and aprotic solvent. The agreement between theoretical and experimental results corroborates our choice for the level of theory employed in this work. It is also interesting to note that the 1CHU $S_{1}$${}^{1}\left(\pi_{5}\pi_{6}^{*}\right)$ excited state is red-shifted in comparison to the value computed for uracil, which can be rationalized remembering that electron donor and/or hyperconjugative substituents in $N_{1}$ position uracil decreases the HOMO–LUMO energy gap [41].

The $S_{2}$${}^{1}\left(n_{O}\pi_{6}^{*}\right)$ state is the second singlet excited state, with a wavefunction best described, in relation to the ground state, by a single electronic transition from the $n_{O}$ non-bonding orbital, localized on the $O_{8}$ oxygen, to the $\pi_{6}^{*}$ orbital. The $S_{2}$${}^{1}\left(n_{O}\pi_{6}^{*}\right)$ state is at 4.84 eV (μ=2.06 Debye) vertically above the ground state, with the corresponding electronic transition with a nearly zero oscillator strength. TD-PBE0/6-31++G(d,p) calculations [34] foresee the ${}^{1}\left(n_{O}\pi^{*}\right)$ state as being the $S_{1}$ state, lying 4.8 eV vertically above the ground state, while the $S_{2}$ state is predicted to be a ${}^{1}\left(\pi \pi^{*}\right)$ state, that is, a reverse sequence predicted by us at the MS-CASPT2(12,9)/cc-pVDZ level of theory. It is also interesting to note that for uracil canonical nucleobase, the $S_{1}$ excited state is predicted [39] to be a ${}^{1}\left(n_{O}\pi_{6}^{*}\right)$ state 4.93 eV vertically above the ground state minimum, and the $S_{2}$ state is a ${}^{1}\left(\pi \pi^{*}\right)$5.18 eV above the ground state in the Franck–Condon region, being the electronic transition from the ground state associated with an oscillator strength of 0.195. Furthermore, the $\pi \to \pi^{*}$ bright state of 1CHU is red-shifted by 0.47 eV in relation to the energetic position observed in canonical uracil nucleobase (Table 2), which fits well with the experimental red-shift of 0.4 eV from uracil to 1CHU [21,42].

As it can be noticed in Table 2, the other singlet excited states are computed to be at least 1.08 eV higher in energy than the S${}_{2}$ excited state. As it has been observed experimentally [34], they do not play a relevant role in the photophysical deactivation pathways of 1CHU. Therefore, we will not put emphasis on them.

As for the triplet excited electronic states (Table 2), the lowest one is the $T_{1}$${}^{3}\left(\pi_{5}\pi_{6}^{*}\right)$ state, 3.73 eV vertically above the ground state at the Franck–Condon (FC) region, with a dipole moment of 3.49 Debye. This electronic state is the triplet analogue of the $S_{1}$${}^{1}\left(\pi_{5}\pi_{6}^{*}\right)$ state, or in other words, it is described by a singly excited configuration from the $\pi_{5}$ to the $\pi_{6}^{*}$ orbital. The next triplet excited state, the $T_{2}$${}^{3}\left(n_{O}\pi_{6}^{*}\right)$, is derived from the ground state by a $n_{O}\to \pi_{6}^{*}$ single excitation, computed to be 4.72 eV (μ=2.10 Debye) vertically above the ground state minimum structure. The $T_{2}$${}^{3}\left(n_{O}\pi_{6}^{*}\right)$ state is in the same energy region as the $S_{1}$${}^{1}\left(\pi_{5}\pi_{6}^{*}\right)$ (4.71 eV). TD-PBE0/6-31++G(d,p) results [34] report the same energetic ordering for the triplet states as that obtained by us at the MS-CASPT2(12,9)/cc-pVDZ level of theory; however, the TD-PBE0/6-31++G(d,p) energies are red-shifted by about 0.3 eV in comparison with our present results. It is also noteworthy that uracil [39] displays the same pattern for the triplet states as that observed for 1CHU.

The last triplet excited state calculated by us is the $T_{3}$${}^{3}\left(\pi_{3}\pi_{6}^{*}\right)$ (μ=2.16 Debye) (Table 2), placed vertically 5.45 above the ground-state minimum and derived from the ground-state by the $\pi_{3}\to \pi_{6}^{*}$ one-electron promotion. As can be seen in Table 2, the $T_{3}$${}^{3}\left(\pi_{3}\pi_{6}^{*}\right)$ state is in a higher energy region, which hinders its participation in the main excited states deactivation pathways of 1CHU.

The same energetic order and nature of the electronic states observed in vacuum is reproduced in water (Table 2), with a small hypsochromic shift observed for the $S_{2}$${}^{1}\left(n_{O}\pi_{6}^{*}\right)$, $T_{1}$${}^{3}\left(\pi_{5}\pi_{6}^{*}\right)$, and $T_{2}$${}^{3}\left(n_{O}\pi_{6}^{*}\right)$ states. This destabilization can be attributed to their smaller dipole moment compared to the ground state, suggesting that the solvent effects are less pronounced in these electronic states. Regarding the gas phase, a bathochromic shift of 0.05 eV is predicted for $S_{1}$${}^{3}\left(\pi_{5}\pi_{6}^{*}\right)$, as it is observed experimentally when going from ethyl acetate to water [21] solution.

### 2.2. Stationary Excited States and Minimum Energy Crossing Points

A mandatory step to explain the nonadiabatic photophysics and deactivation processes of 1CHU consists of searching the most important excited state critical points. The $S_{2}$${}^{1}{\left(n_{O}\pi_{6}^{*}\right)}_{\min}$ optimized structure is adiabatically 4.22 eV above the ground state minimum structure. In comparison to the ground state, the pyrimidine ring remains planar (Appendix A), but the $C_{4}O_{8}$ bond is elongated by about 0.128 Å, due to the electron transfer from the non-bonding orbital localized on the oxygen atom to an anti-bonding orbital in the pyrimidine ring. In addition, the $C_{4}C_{5}$ bond is shortened by 0.084 Å. As can be seen in Table 1, the computed pyrimidine ring distances observed for 1CHU agree well with those reported for the minimum structure of the uracil ${}^{1}\left(n_{O}\pi^{*}\right)$ excited state [40].

The $T_{1}$${}^{3}{\left(\pi \pi^{*}\right)}_{\min}$ was the last structure optimized, found 3.19 eV adiabatically above the ground state minimum. In relation to the ground state structure, the most prominent change is a stretch of 0.128 Å on the $C_{5}C_{6}$ bond. This structure is best described by a boat-like conformation (${}^{3,6}B$) (Appendix A) according to Cremer–Pople [43] and Boyens [44] parameters (Q=0.45 Å, $\Phi =62^{\circ }$, and $\Theta =78^{\circ }$).

The first minimum energy crossing point (MECP) optimized was the ($S_{0}$/$S_{1}$)${}_{\mathrm{SSCP}}$ structure, between the $S_{0}$ and $S_{1}$${}^{1}\left(\pi_{5}\pi_{6}^{*}\right)$ singlet electronic states, placed adiabatically 3.97 eV above the ground-state minimum structure. At the ($S_{0}$/$S_{1}$)${}_{\mathrm{SSCP}}$ optimized structure, the energy gap between the electronic $S_{0}$ and $S_{1}$ states is computed to be 0.11 eV. Single-point energy calculation with this optimized structure, carried out at the MS(3S+3T)-CASPT2(12,9)/cc-pVDZ level of theory, reveals that in the same energetic region we also have a three-state crossing point involving the $S_{0}$, $S_{1}$${}^{1}\left(\pi_{5}\pi_{6}^{*}\right)$, and $T_{1}$${}^{3}\left(\pi_{5}\pi_{6}^{*}\right)$ states. For uracil, it is interesting to note that a singlet-triplet crossing point between the $T_{1}$${\;}^{3}\left(\pi \pi^{*}\right)$ and the $S_{1}$${\;}^{1}\left(\pi \pi^{*}\right)$ states was found [39] in the vicinity of the ($S_{0}$/$S_{1}$)${}_{\mathrm{SSCP}}$, 4.2 eV adiabatically above the ground-state minimum. It is worth mentioning that the search for a minimum on the $S_{1}$${\;}^{1}\left(\pi_{6}\pi_{5}^{*}\right)$ PES leads directly to the ($S_{0}$/$S_{1}$)${}_{\mathrm{SSCP}}$ singlet–singlet minimum energy crossing structure between the ground and $S_{1}$${}^{1}\left(\pi_{5}\pi_{6}^{*}\right)$ states.

In comparison to the ground state, the ($S_{0}$/$S_{1}$)${}_{\mathrm{SSCP}}$ structure exhibits a stretched (0.107 Å) bond and twisted $HC_{5}C_{6}H$ dihedral (d${}_{HC_{5}C_{6}H}∼115^{\circ }$) angle, similar to the so-called ethylenic ${}^{1}\pi \pi^{*}$/GS conical intersection (Appendix A). Ring conformation analysis classifies the 1CHU ($S_{0}$/$S_{1}$)${}_{\mathrm{SSCP}}$ geometry in a Boeyens group differently from that reported for uracil [40] (${}^{6}T_{2}$ for 1CHU and ${}^{3,6}B$ for uracil). Nevertheless, the Cremer and Pople parameters are close to each other (Q=0.48 Å, $\Phi =98^{\circ }$, $\Theta =100^{\circ }$ for 1CHU and Q=0.50 Å, $\Phi =65^{\circ }$, and $\Theta =87^{\circ }$ for uracil).

Two other minimum energy crossing points were optimized; one of them is the singlet-triplet crossing point between the $S_{1}$${}^{1}\left(\pi_{5}\pi_{6}^{*}\right)$ and $T_{2}$${}^{3}\left(n_{O}\pi_{6}^{*}\right)$ states (($S_{1}$/$T_{2}$)${}_{\mathrm{STCP}}$). Again, single-point energy calculations using this optimized structure as reference place the $S_{2}$${}^{1}\left(n_{O}\pi_{6}^{*}\right)$ in the same energetic region, which is 0.33 eV above the $S_{2}$${\;}^{1}{\left(n_{O}\pi^{*}\right)}_{\min}$ minimum energy structure. This structure differs slightly from the ground state minimum, except for the $N_{4}C_{4}$ bond, which becomes 0.05 Å longer. The other MECP involves the $T_{1}$ and $T_{2}$ electronic states (($T_{1}$/$T_{2}$)${}_{\mathrm{TTCP}}$). Again, vertical energy calculations evidences that the $S_{1}$ state is in the same energetic region. Its most structural striking feature is the elongation of the $C_{4}O_{8}$ bond by 0.158 Å, in comparison with the distance observed in the ground state.

The structures mentioned above were re-optimized in water by employing the PCM model. The optimized parameters in vacuum and water (Table 1) are very close to each other, as can be seen by the superposition of the optimized geometries in both environments (Appendix A). The most significant bond length deviation is about 0.01 Å, which means that the effects of water have a minor influence on the molecular structures. The same conclusion can be reached by comparing the adiabatic excitation energies computed in vacuum and water (Table 3). Nevertheless, water solvation effects are more pronounced on the MECPs structures (Appendix A), with the most striking alteration observed on the $N_{1}C_{2}$ bond, which for the ($S_{0}$/$S_{1}$)${}_{\mathrm{SSCP}}$ is shorter by about 0.37 Å in water.

If water solvation effects do not change the structural parameters much, it has a noticeable effect on the topology of the $S_{1}$${}^{1}\left(\pi_{5}\pi_{6}^{*}\right)$ PES. Unlike in vacuum, in water, we observed a minimum on the $S_{1}$${}^{1}\left(\pi_{5}\pi_{6}^{*}\right)$ PES, with a boat conformation (Q=0.40 Å, $\Phi =63^{\circ }$, and $\Theta =90^{\circ }$) (Appendix A) placed adiabatically 4.16 eV above the ground-state minimum structure. A similar boat-like structure was already reported for uracil at the PCM/TD-DFT(PBE0) level [41], although other authors assigned a planar structure to this minimum [45]. A final remark is that vertical excitation energies calculations in water have not shown a third state degenerated with the optimized MECP structures.

### 2.3. Excited State Deactivation Pathways

Before discussing in detail the main photophysical events, an overall picture of the whole process is displayed in Figure 2 in order to ease the reading of our discussion.

The next step to elucidate the relevant deactivation mechanisms concerns in describing the probable pathways by connecting the relevant regions computed previously. To this end, we first performed linear interpolation in internal coordinates, defined via Z-Matrix, connecting the $S_{1}$${}^{1}\left(\pi_{5}\pi_{6}^{*}\right)$ structure from the Franck–Condon region to the ($S_{0}$/$S_{1}$)${}_{\mathrm{SSCP}}$ region, where the spin-orbit coupling (SOC) is ∼2$\;{\mathrm{cm}}^{-1}$. As mentioned before, at each LIIC point, we carried out an MS(3S+3T)-CASPT2/cc-pVDZ single-point energy calculation to have a global view of the possible events. The computed LIIC scan is displayed by PATH I in Figure 3. It can be noticed that after the $S_{1}$${}^{1}\left(\pi_{5}\pi_{6}^{*}\right)$ state is initially populated, it evolves barrierlessly to the ($S_{0}$/$S_{1}$)${}_{\mathrm{SSCP}}$ region, 0.74 eV adiabatically below the starting point. Once in the three-state crossing region, two possible deactivation pathways can be seen: (i) an efficient ultrafast and radiationless decay to the ground state via internal conversion process or (ii) the $T_{1}$${}^{3}\left(\pi_{5}\pi_{6}^{*}\right)$ state can be populated via an intersystem crossing process. As the SOC computed in the three-state crossing region is small, the ultrafast decay toward the ground state via the minimum energy crossing point is more likely than the triplet state population.

A second route exploited by us is the direct population transfer from the $S_{1}$${}^{1}\left(\pi_{5}\pi_{6}^{*}\right)$ state to the triplet states manifold, also investigated by means of LIIC connecting the structures from the Franck-Condon region and other characteristic regions. As displayed in Path II (Figure 3), the ($S_{1}$/$T_{2}$)${}_{\mathrm{STCP}}$ region can be reached barrierlessly from the Franck-Condon region. Due to the sizable SOC between the $S_{1}$${\;}^{1}\left(\pi_{5}\pi_{6}^{*}\right)$ and $T_{2}$${\;}^{3}\left(n_{O}\pi_{6}^{*}\right)$ states at this region ($19\;{\mathrm{cm}}^{-1}$) and to the small energy gap between the electronic states (0.003 eV), the $T_{2}$${\;}^{3}\left(n_{O}\pi_{6}^{*}\right)$ state can be populated. In comparison to uracil, Climent and coworkers suggested that the ${}^{1}\left(\pi \pi^{*}\right)$ state could also be a doorway to populate the triplet states manifold of uracil [39], for which the SOC between $S_{1}$ and $T_{2}$ is computed to be $25\;{\mathrm{cm}}^{-1}$.

After population transfer from the $S_{1}$ to the $T_{2}$ state (Path II), an extra-energy of 0.39 eV is released to access the minimum of the T${}_{2}$ state. LIIC scans evidence that it is a shallow region, since the ($T_{2}$/$T_{1}$)${}_{\mathrm{TTCP}}$ structure can be easily accessible (0.20 eV above the $T_{2}$${}^{3}{\left(n_{O}\pi_{6}^{*}\right)}_{\min}$) (Path III), from where the system evolves to the minimum of $T_{1}$ potential energy hypersurface dissipating 1.13 eV of energy.

The photophysical deactivation mechanisms of 1CHU in water were also investigated by us employing the PCM model. An LIIC interpolated pathway connecting the Franck–Condon and $S_{1}$${\;}^{1}{\left(\pi_{5}\pi_{6}^{*}\right)}_{\min}$ regions (Figure 3, PATH IV) indicates that the system evolves via a barrierless path along its potential energy hypersurface towards the $S_{1}$${\;}^{1}{\left(\pi_{5}\pi_{6}^{*}\right)}_{\min}$ minimum energy region, 0.55 eV adiabatically below the Franck–Condon region. From the $S_{1}$${\;}^{1}{\left(\pi_{5}\pi_{6}^{*}\right)}_{\min}$ region, the system can evolve towards the ($S_{0}$/$S_{1}$)${}_{\mathrm{SSCP}}$ singlet-singlet minimum energy crossing point, where it returns to the ground state via this fast and radiationless photophysical path. It is interesting to note that the ($S_{0}$/$S_{1}$)${}_{\mathrm{SSCP}}$ is located at about the same energetic region as the $S_{1}$${\;}^{1}{\left(\pi_{5}\pi_{6}^{*}\right)}_{\min}$ structure, at 4.07 eV adiabatically above the ground state optimized structure. To investigate the accessibility of the ($S_{0}$/$S_{1}$)${}_{\mathrm{SSCP}}$ singlet–singlet minimum energy crossing point from the $S_{1}$${\;}^{1}{\left(\pi_{5}\pi_{6}^{*}\right)}_{\min}$ structure region, we connect the points by using linear interpolation in internal coordinates. However, as the interpolated pathway represents an upper limit to the real path, we optimized the transition state structure (TS) between the initial and final points at the SA(3S)-CASSCF/cc-pVDZ level of theory, because this kind of calculation at the MS-CASPT2 level is too computationally demanding; it is important to mention that the final energy of the TS was computed at the MS(3S+3T)-CASPT2/cc-pVDZ level. The computed energetic barrier employing the optimized TS structure is about 0.18 eV, while that along the computed LIIC pathway is 0.24 eV. Therefore, due to the small energetic barrier (0.18 eV), the evolution from the $S_{1}$${\;}^{1}{\left(\pi_{5}\pi_{6}^{*}\right)}_{\min}$ region to the crossing region with the ground state, ($S_{0}$/$S_{1}$)${}_{\mathrm{SSCP}}$, is a very probable deactivation path. It is interesting to note that the presence of a minimum on the $S_{1}$${\;}^{1}\left(\pi_{5}\pi_{6}^{*}\right)$ potential energy hypersurface and an energetic barrier along the path towards the (($S_{0}$/$S_{1}$)${}_{\mathrm{SSCP}}$ region can trap the population on the minimum region, and a fraction of the energy could be released by, albeit weak, fluorescence (3.67 eV).

The photophysical mechanisms behind the population of the triplet state (Figure 3, PATH V) were also investigated in water. The singlet–triplet minimum energy crossing point between the $S_{1}$${}^{1}\left(\pi_{5}\pi_{6}^{*}\right)$ and $T_{2}$${}^{3}\left(n_{O}\pi_{6}^{*}\right)$ states, ($S_{1}$/$T_{2}$)${}_{\mathrm{STCP}}$, is computed to be adiabatically 0.23 eV higher in energy than the $S_{1}$${\;}^{1}{\left(\pi_{5}\pi_{6}^{*}\right)}_{\min}$ region, with a SOC $∼18\;{\mathrm{cm}}^{-1}$, at about the same value computed in the gas phase. From the ($S_{1}$/$T_{2}$)${}_{\mathrm{STCP}}$ region, the system can evolve to the $T_{2}$${\;}^{3}{\left(n_{O}\pi_{6}^{*}\right)}_{\min}$ state minimum region, dissipating an energy of 0.18 eV. Nonetheless, as displayed in PATH VI (Figure 3), the population can also be transferred from the $T_{2}$${}^{3}\left(n_{O}\pi_{6}^{*}\right)$ to the $T_{1}$${}^{3}\left(\pi_{5}\pi_{6}^{*}\right)$ state by means of an internal conversion mediated by the ($T_{2}$/$T_{1}$)${}_{\mathrm{TTCP}}$ triplet–triplet minimum energy crossing point, located 0.22 eV adiabatically above the $T_{2}$${\;}^{3}{\left(n_{O}\pi_{6}^{*}\right)}_{\min}$ region.

In short, two deactivation mechanisms can be foreseen for 1CHU. One of them is the ultrafast and nonradiative decay to the ground state, similar to that observed in uracil. This result is in line with experimental findings obtained in ethyl acetate solution, in which a high internal conversion yield to the ground state (46%) was observed. The other relaxation pathway involves the direct population transfer from the S${}_{1}{\;}^{1}\left(\pi_{5}\pi_{6}^{*}\right)$ state to the T${}_{2}{\;}^{3}\left(n_{O}\pi_{6}^{*}\right)$ state via an ISC process. A high ISC yield (54%) was also experimentally found; however, as emphasized by the authors, this value is overestimated since a residual contribution associated with a long-lived single dark state population could not be disentangled. Experimental findings were analyzed by comparing their results with those obtained for uracil with the multireference configuration interaction (MRCI) method [46], which suggests the ${}^{1}\left(n\pi^{*}\right)$ as being the lowest-lying singlet excited state. It is important to mention that our results do not suggest that the dark ${}^{1}\left(n_{O}\pi_{6}^{*}\right)$ singlet state can be populated, because according to our results, the lowest-lying singlet excited state is the ${}^{1}\left(\pi_{5}\pi_{6}^{*}\right)$ state, with the ${}^{1}\left(n_{O}\pi_{6}^{*}\right)$ state always higher in energy. Furthermore, test calculations on the FC region indicate that the relative position the ${}^{1}\left(n_{O}\pi^{*}\right)$ and ${}^{1}\left(\pi \pi^{*}\right)$ depend on the substituent attached to the $N_{1}$ position (see Appendix A), as discussed before. Experiments carried out in water indicate that after photoexcitation, 60% of the population returns to the ground-state and only 3% is associated with the triplet states population. The difference between the results obtained in ethyl acetate and water can be rationalized considering the presence of a minimum on the S${}_{1}$ hypersurface observed in water, where the population can be trapped, diminishing the fraction of population that follows towards the singlet-triplet crossing region. Another reason is the competition between the internal conversion and intersystem crossing processes. As we do not obtain a substantial increment in the SOC computed in water, we conclude that the internal conversion should be the dominant mechanism. Unfortunately, to conclude which one is the most likely mechanism, a quantitative analysis of the deactivation pathways can only be obtained by carrying out nonadiabatic dynamics simulations, which is out of the scope of our result.

## 3. Materials and Methods

Characteristic points along the potential energy hypersurfaces, for instance, ground and excited states minima and minimum energy crossing points, were optimized with the Multi-State CASPT2 (MS-CASPT2, Multi-State Complete Active-Space Second-Order Perturbation Theory) [47] based on a zeroth-order wave function computed with the state-averaged complete active space self-consistent field (SA-CASSCF) [48] method. All calculations were carried out with the double-ζ atomic basis set (cc-pVDZ) [49], and integrals were computed employing the RIJK approximation [50] to speed up the integral calculations. The MS-CASPT2 calculations were performed with the standard zeroth-order Hamiltonian [51], freezing core orbitals, without applying the IPEA (Ionization Potential-Electron Affinity Shift) shift correction [52], and with an imaginary level-shift [53] of 0.2 a.u. to deal with intruder state problems.

Minimum energy crossing-points (MECPs) were optimized as the lowest energy point obtained with the restricted Lagrange multipliers technique [54], imposing the constraint of degeneracy between the two states considered. SSCP was be used for singlet–singlet MECPs, TTCP for triplet–triplet minimum energy crossing points, and STCP for singlet–triplet MECPs. Characteristic points were connected by means of linear interpolation in internal coordinates defined via Z-Matrix (LIIC), since minimum energy path calculations at the MS-CASPT2/cc-pVDZ level of theory are currently unfeasible due to the size of the system. It is worth recalling that LIICs usually provide a reasonable representation of the real energy hypersurface, which allows a qualitative overview of the topology of the real hypersurface. Still, the computed energy barriers along the LIIC pathway are commonly overestimated. Spin-orbit couplings (SOCs) were computed using the Atomic Mean Field Integrals (AMFI) [55,56], as in previous works [57,58]. Water solvation effects were mimicked by applying the Polarizable Continuum Model (PCM) [35]. When applicable, all characteristic points were re-computed accordingly in order to take into account solvation effects.

The employed active space contains twelve electrons distributed over nine orbitals (Figure 4), encompassing a non-bonding orbital localized on the $O_{8}$ position ($n_{O}$) plus eight orbitals described as π and $\pi^{*}$ delocalized on the pyrimidine ring and oxygen atoms (CAS(12,9)). The molecular orbitals localized in the cyclohexyl moiety were not included in the active space, since electronic transitions from these orbitals are not relevant for the photophysics of 1CHU. In addition, the lone pair associated with the $O_{7}$ atom was kept inactive because its average occupation number is always close to two. Vertical excitation energies calculations were performed at the same level of theory and active space, averaging over the three lowest lying singlet and triplet electronic states (SA(3S+3T)-CASSCF), with specific calculations for each spin multiplicity. Test calculations with larger basis sets, active space, and different number of states in the state-averaged procedure yielded results almost identical to that computed at the MS(3S+3T)-CAS(12,9)/cc-pVDZ level of theory; based on that, we decided to perform our calculations at the MS(3S+3T)-CAS(12,9)/cc-pVDZ level of theory (see Appendix A for further details).

Electronic structure calculations were performed with the OpenMolcas [59] suite of program, without imposing spatial symmetry restrictions. The COLUMBUS software [60] was employed to interpolate the geometries for the LIICs calculations.

## 4. Conclusions

In this contribution, we presented a systematic investigation of the photophysical deactivation pathways of 1-cyclohexyluracil in vacuum and water. Critical points on different potential energy hypersurfaces were localized and connected by LIIC scans to verify their accessibility. After the irradiation and population of the lowest S${}_{1}{\;}^{1}\left(\pi \pi^{*}\right)$ bright state, two plausible relaxation mechanisms can take place in both environments. One of them (Route 1) is the internal conversion to the ground state, the same nonradiative decay observed in uracil. Another plausible deactivation mechanism (Route 2) is the transfer of the population to the T${}_{2}{\;}^{3}\left(n_{O}\pi^{*}\right)$ state. As this singlet-triplet crossing point involves electronic states of different characters, we found an SOC around 20 cm${}^{-1}$, which implies that the population transfer from the singlet to the triplet state is possible. This is in line with experimental data that measured similar values of IC and ISC quantum yields in ethyl acetate solution. In water, the suppression of the quantum yield of triplet state population was experimentally noted. According to our theoretical results, this could be associated with the presence of a minimum energy region on the S${}_{1}{\;}^{1}\left(\pi \pi^{*}\right)$ state potential energy surface together with an energetic barrier that makes a little difficult to reach the conical intersection with the ground state, suggesting that a portion of the excess of energy would be released as fluorescence, albeit weak.

## Figures and Tables

**Figure 1 molecules-26-05191-f001:**
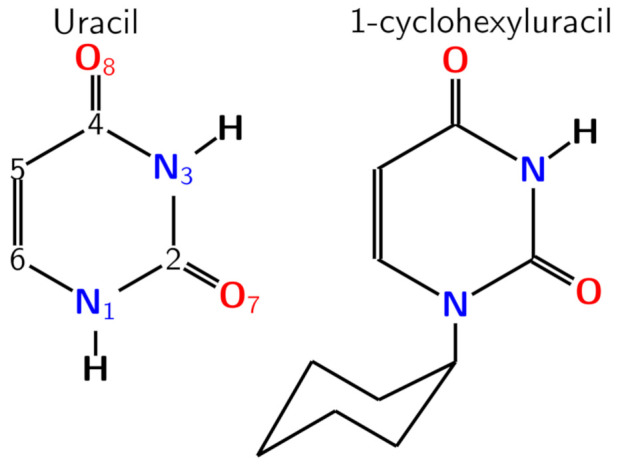
Molecular structure and numeration scheme of uracil and 1-cyclohexyluracil molecules.

**Figure 2 molecules-26-05191-f002:**
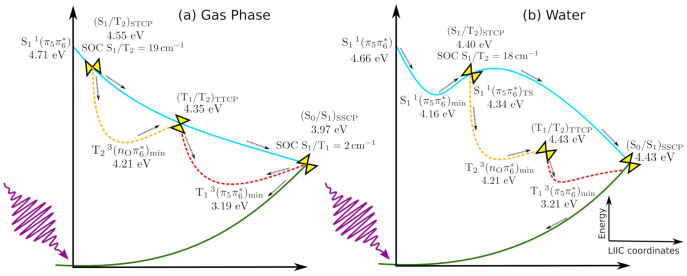
Schematic representation of the main photophysical events computed for 1CHU in (**a**) gas phase and (**b**) water. Energies in relation to the ground state optimized geometry are reported in eV, and the SOC values in cm${}^{-1}$.

**Figure 3 molecules-26-05191-f003:**
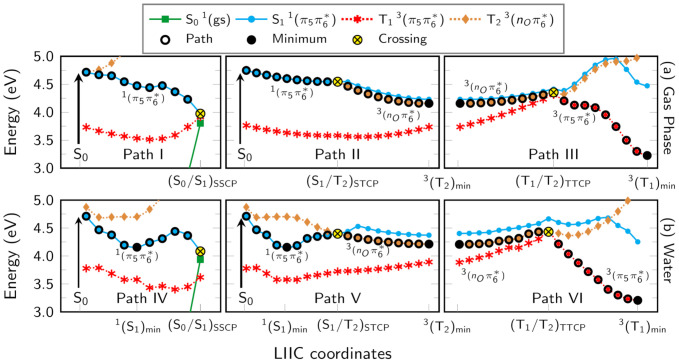
Linear interpolation in internal coordinates defined via Z-Matrix deactivation pathways in (**a**) gas phase and (**b**) water. For each LIIC interpolated structure, a single-point energy calculation at the MS(3S,3T)-CASPT2(12,9)/cc-pVDZ level of theory was carried out.

**Figure 4 molecules-26-05191-f004:**
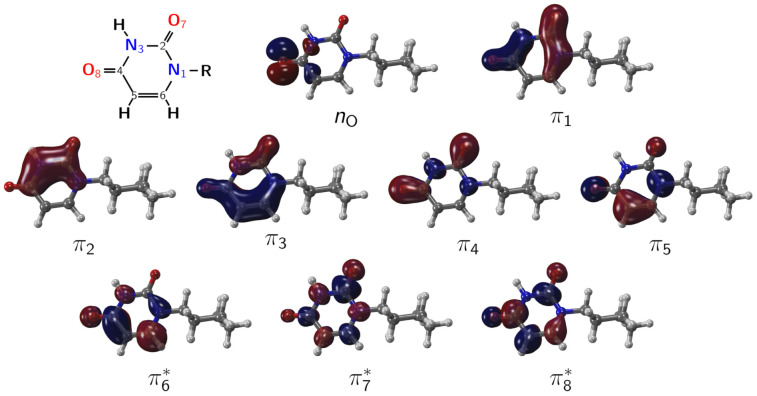
Active space employed during geometry optimizations and vertical excitation energies calculations, encompassing twelve electrons and nine orbitals. The orbitals were obtained in terms of state-averaged valence natural orbitals calculated at the SA(3S+3T)-CASSCF level in the ground-state equilibrium geometry.

**Table 1 molecules-26-05191-t001:** Main atomic distances (in Å) and dihedral angle d${}_{HC_{5}C_{6}H}$ (in degree, ${}^{\circ }$) for 1CHU selected geometries, in gas phase and water (as described by the Polarized Continuum Model, PCM), optimized at the MS-CASPT2(12,9)/cc-pVDZ level of theory. Other theoretical results and corresponding values for uracil are displayed for comparison.

	$N_{1}C_{2}$	$C_{2}N_{3}$	$N_{3}C_{4}$	$C_{4}C_{5}$	$C_{5}C_{6}$	$C_{6}N_{1}$	$C_{2}O_{7}$	$C_{4}O_{8}$	d${}_{HC_{5}C_{6}H}$
	1CHU-Gas Phase								
$S_{0}$ ${}_{\min}$	1.405	1.386	1.409	1.459	1.364	1.383	1.224	1.226	−0.1
$S_{0}$ ${}_{\min}{\;}^{a}$	1.401	1.384	1.409	1.452	1.350	1.375	1.216	1.217	0.0
$S_{2}$ ${}^{1}{\left(n_{O}\pi_{6}^{*}\right)}_{\min}$	1.386	1.400	1.397	1.375	1.413	1.410	1.227	1.354	0.1
$T_{1}$ ${}^{3}{\left(\pi_{5}\pi_{6}^{*}\right)}_{\min}$	1.402	1.401	1.411	1.448	1.492	1.395	1.225	1.224	12.1
$T_{2}$ ${}^{3}{\left(n_{O}\pi_{6}^{*}\right)}_{\min}$	1.387	1.402	1.396	1.379	1.408	1.412	1.223	1.345	−2.5
($S_{0}$/$S_{1}$)${}_{\mathrm{SSCP}}$	1.453	1.387	1.443	1.478	1.471	1.354	1.213	1.221	−115.5
($S_{1}$/$T_{2}$)${}_{\mathrm{STCP}}$	1.425	1.369	1.458	1.429	1.391	1.415	1.236	1.262	3.3
($T_{1}$/$T_{2}$)${}_{\mathrm{TTCP}}$	1.367	1.416	1.402	1.381	1.366	1.432	1.225	1.384	−0.6
	**1CHU-Water**								
$S_{0}$ ${}_{\min}$	1.398	1.384	1.403	1.452	1.370	1.379	1.230	1.236	−0.1
$S_{1}$ ${}^{1}{\left(\pi_{5}\pi_{6}^{*}\right)}_{\min}$	1.449	1.363	1.480	1.414	1.496	1.354	1.238	1.213	9.4
$S_{2}$ ${}^{1}{\left(n_{O}\pi_{6}^{*}\right)}_{\min}$	1.380	1.400	1.398	1.374	1.412	1.412	1.233	1.359	−3.2
$T_{1}$ ${}^{3}{\left(\pi_{5}\pi_{6}^{*}\right)}_{\min}$	1.394	1.401	1.405	1.441	1.497	1.398	1.228	1.231	8.7
$T_{2}$ ${}^{3}{\left(n_{O}\pi_{6}^{*}\right)}_{\min}$	1.383	1.394	1.399	1.382	1.407	1.415	1.233	1.352	0.9
($S_{0}$/$S_{1}$)${}_{\mathrm{SSCP}}$	1.415	1.408	1.410	1.455	1.475	1.390	1.223	1.244	−118.1
($S_{1}$/$T_{2}$)${}_{\mathrm{STCP}}$	1.401	1.375	1.434	1.431	1.375	1.422	1.236	1.287	0.8
($T_{1}$/$T_{2}$)${}_{\mathrm{TTCP}}$	1.368	1.415	1.402	1.380	1.367	1.432	1.225	1.383	−0.6
	**Uracil-Gas Phase**								
$S_{0}$ ${}_{\min}$	1.394	1.388	1.417	1.465	1.361	1.381	1.220	1.221	0.0
${}^{1}{\left(n_{O}\pi^{*}\right)}_{\min}{\;}^{b}$	1.377	1.399	1.399	1.375	1.416	1.406	1.219	1.355	−0.9
($S_{0}$/$S_{1}$)${}_{\mathrm{SSCP}}{\;}^{b}$	1.447	1.379	1.465	1.426	1.483	1.347	1.214	1.232	136.6

${}^{a}$ Brister and Crespo-Hernández [34] at the B3LYP/6-11++G(p,d) level of theory. ${}^{b}$ Yamazaki and Taketsugu [40] at the MS-CASPT2/Sapporo-DZP level of theory.

**Table 2 molecules-26-05191-t002:** Vertical excitation energies (ΔE, eV), oscillator strength (*f*), and dipole moment (μ, Debye) for 1CHU in gas phase and water obtained at the MS(3S+3T)-CASPT2(12,9)/cc-pVDZ level of theory at the Franck–Condon region. Solvents effects were taken into account with the Polarizable Continuum Model (PCM). Uracil ground state geometry optimization and the corresponding vertical excitation energies were obtained at the same level of theory employed for 1CHU. Experimental absorption peaks (Exp.), obtained in ethyl acetate and aqueous solutions, together with previous theoretical results reported for 1CHU, are also displayed for comparison.

	Gas Phase							Water			
	1CHU					Uracil		1CHU			
	ΔE	*f*	μ	ΔE${}^{a}$	Exp. ${}^{b}$	ΔE	Exp. ${}^{C}$	ΔE	*f*	μ	Exp. ${}^{d}$
$S_{0}$ ${}^{1}\left(gS\right)$	0.00	—	4.56	0.0	—	0.00	—	0.00	—	5.78	—
$T_{1}$ ${}^{3}\left(\pi_{5}\pi_{6}^{*}\right)$	3.73		3.49	3.3		3.80		3.80		4.28	
$S_{1}$ ${}^{1}\left(\pi_{5}\pi_{6}^{*}\right)$	4.71	0.360	7.27	5.1	4.68	5.18	5.08	4.66	0.366	8.44	4.61
$T_{2}$ ${}^{3}\left(n_{O}\pi_{6}^{*}\right)$	4.72		2.10	4.4		4.71		4.89		2.94	
$S_{2}$ ${}^{1}\left(n_{O}\pi_{6}^{*}\right)$	4.84	0.001	2.06	4.8		4.93		5.04	0.001	3.07	3.23
$T_{3}$ ${}^{3}\left(\pi_{3}\pi_{6}^{*}\right)$	5.45		2.16	4.7		5.33		5.53		2.95	

${}^{a}$ Brister and Crespo-Hernández [34] at level the TD-PBE0/6-31++G(d,p) level of theory. ${}^{b}$ Hare et al. [21] in ethyl acetate solution. ${}^{c}$ Clark et al. [42] in vacuum. ${}^{d}$ Hare et al. [21] in water.

**Table 3 molecules-26-05191-t003:** Adiabatic energies (eV) relative to the ground state minimum in vacuum and water.

Medium	$S_{1}$ ${\;}^{1}{\left(\pi_{5}\pi_{6}^{*}\right)}_{\min}$	$S_{2}$ ${\;}^{1}{\left(n_{O}\pi_{6}^{*}\right)}_{\min}$	$T_{1}$ ${\;}^{3}{\left(\pi_{5}\pi_{6}^{*}\right)}_{\min}$	$T_{2}$ ${\;}^{3}{\left(n_{O}\pi_{6}^{*}\right)}_{\min}$	($S_{0}/S_{1}$)${}_{\mathrm{SSCP}}$	($S_{1}/T_{2}$)${}_{\mathrm{STCP}}$	($T_{1}/T_{2}$)${}_{\mathrm{TTCP}}$
Vacuum	—	4.22	3.19	4.16	3.97	4.55	4.35
Water	4.16	4.57	3.21	4.21	4.08	4.40	4.43

## Data Availability

The data presented in this study can be made available upon request to the corresponding author. See also Appendix A.

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
