# Peer review of "Photophysical Deactivation Mechanisms of the Pyrimidine Analogue 1-Cyclohexyluracil"

_molecules, 2021, doi:10.3390/molecules26175191_

Round 1

Reviewer 1 Report

The authors present a computational investigation on the electronic potential energy surfaces of 1CHU, using state of the art quantum chemistry methods. The aim is to assess the most likely decay pathway in the gas phase and in solution. The topic is interesting, but before publication the authors should consider the following remarks.

1) I suggest the authors to give another careful read to the abstract, which contains several errors. Moreover, I do not understand what are the "ε" values reported in page 2, just below figure 1. Usually, the Greek letter epsilon is used to indicate the (relative) dielectric constant, but 0.03 cannot be the relative dielectric constant of water. Please clarify.

2) In the paragraph devoted to the discussion of the possible deactivation channels (last paragraph of section 2.3), the authors report, in my opinion, an incorrect account of the experimental results of ref. 21. In fact, what is said in ref. 21 is that in ethyl acetate solution the 54% of 1CHU molecules showing a long decay time are not necessarily all in a triplet state: the dark S1 state is also populated in that case, and its decay time is long enough to make it difficult to experimentally disentangle it from the triplet. The authors should re-read ref. 21 and reformulate their discussion.

3) The authors show that, at variance with gas phase, in water solution the S1 state of 1CHU presents a minimum, and they computed by LIIC an energy barrier of 0.24 eV connecting the S1 minimum and the S1/S0 CI. However, given the importance of that barrier in the decay dynamics, the author should try to determine it in a proper way (for example, by performing a minimum energy path scan, or a search for a transition state connecting the S1 minimum and the S1/S0 CI in water).

4) According to authors, 1CHU should show a non negligible fluorescence quantum yield in water. Are there experimental evidences of that? Actually, it seems to be quite unlikely that radiative decay is important in 1CHU, given the vanishing oscillator strength of the S1state. Please clarify.

5) In photochemistry, "intersystem crossing" designate a nonradiative process, not a topological feature. So, it should not be employed to name a singlet/triplet crossing point.

Reviewer 2 Report

This work by Valverde and co-workers is a purely theoretical investigation on the photochemistry of 1-cyclohexylyracil (1CHU) molecule. This work is of limited interest to readers as this molecule is very similar to the uracil molecule which has been well studied by others. The authors employed high-level wavefunction based theory, however, the presentation of the results is poor and difficult to follow. The authors seem to have serious issues in understanding the basic concepts in photochemistry. This work would require a significant improvement before I can recommend it for publication. 

Major issues include:

1) The abstract

The author mentioned "internal coversion", the question is how about "fluorescence"?

What does intersystem crossing from T_2^3 state mean? Is this correct?

What does "most population return to ground state via conical intersection" mean? 

Please make sure that the authors really understand what these sentences mean.

2) In the introduction part, the authors did not explain why this 1CHU molecule is important for them to investigate. Is it simply because there were some experimental studies before? Please explain why we should care about this uracil derivative.

3) Should MS-CASPT2 be expanded as multi-state...?

4) What is "canonical counterpart" ? Is this uracil molecule?

5) In Table 2, the authors put the excitation energy in "ethyl acetate" together with that in gas phase for comparison, this does not make sense.

In Table 2, the excitation energy data for 1CHU in water seems to be very inaccurate with the error up to 1 eV and 1.7 eV, which are quite large. In this regard, the calculation results for 1CHU in water throughout the whole paper are unreliable.

6) The presentation of energy curves in Figure 2 is bad and very hard to follow. The readers would need an overall picture of the whole process and then a detailed picture like figure 2. 

The color scheme for T1 and T2 is also not good because two colors are too close. 

7) The authors employed the so-called LIIC technique, however, it's not clear how which internal coordinates are selected. 

8) The statement about the relationship between MECP and conical intersecction seems wrong. 

Round 2

Reviewer 1 Report

With the changes provided by the authors, the manuscript is suitable for publication.

Author Response

Reviewer 1 accepted the revised version of our manuscript, he/she did not ask any further question.

Reviewer 2 Report

This reviewer agrees with the author's reply and is convinced by the revision. However, Figures 2 and 3 should switch their positions in order to give the overall picture first and then detailed description. 

Author Response

Referee Comment: This reviewer agrees with the author’s reply and is convinced by the revision. However, Figures 2 and 3 should switch their positions in order to give the overall picture first and then detailed description.

Author Response: The manuscript was modified accordingly. See Section 2.3, page 6, first paragraph.